# Motor Endplate—Anatomical, Functional, and Molecular Concepts in the Historical Perspective

**DOI:** 10.3390/cells8050387

**Published:** 2019-04-27

**Authors:** Rüdiger Rudolf, Muzamil Majid Khan, Veit Witzemann

**Affiliations:** 1Institute of Molecular and Cell Biology, Mannheim University of Applied Sciences, 68163 Mannheim, Germany; 2Interdisciplinary Center for Neuroscience, Heidelberg University, 69120 Heidelberg, Germany; 3Institute of Toxicology and Genetics, Karlsruhe Institute of Technology, 76344 Eggenstein-Leopoldshafen, Germany; 4Cell Biology and Biophysics, European Molecular Biology Laboratory, 69117 Heidelberg, Germany; muzamil.m.khan@embl.de; 5Max Planck Institute for Medical Research, 69120 Heidelberg, Germany; veit.witzemann@mpimf-heidelberg.mpg.de

**Keywords:** history, motor endplate, neuromuscular junction, skeletal muscle, sympathetic innervation, synapse

## Abstract

By mediating voluntary muscle movement, vertebrate neuromuscular junctions (NMJ) play an extraordinarily important role in physiology. While the significance of the nerve-muscle connectivity was already conceived almost 2000 years back, the precise cell and molecular biology of the NMJ have been revealed in a series of fascinating research activities that started around 180 years ago and that continues. In all this time, NMJ research has led to fundamentally new concepts of cell biology, and has triggered groundbreaking advancements in technologies. This review tries to sketch major lines of thought and concepts on NMJ in their historical perspective, in particular with respect to anatomy, function, and molecular components. Furthermore, along these lines, it emphasizes the mutual benefit between science and technology, where one drives the other. Finally, we speculate on potential major future directions for studies on NMJ in these fields.


*“These were the happy days, for which the current generation might envy us, when the first good, handy, and affordable microscopes came from the workshops of Plössl in Vienna and Pistor and Schick in Berlin, the happy days, when it was still possible to make fundamental discoveries by simply scraping with the blade of a scalpel or the fingernails on an animal’s membrane.”*
*Jacob Henle, Orbituary for Theodor Schwann (1882)*

## 1. Introduction

Fast, nervous-system-controlled voluntary body movement by muscle activity is a principal characteristic of most animals, from simple Cnidaria to the intricate chordates, and man. The complexity of the neuro-muscular system shows a clear dependency on the life style of an animal. Indeed, while free-living and actively moving animals such as fish, insects, or mammals, have well developed nerve and muscle systems, sessile animal life forms such as sponges, bryozoans, or adult sea squirts are either completely devoid or display very simple versions of those. Although the principal concept of brain and nerves controlling voluntary movement was already described in the second century AD by Galen [1,2], first accounts on the cellular contact sites between motor neurons and muscle fibers, which we now call neuromuscular junctions (NMJs) or motor endplates, come from the nineteenth century. Indeed, this coincided with the invention of novel microscopic technologies including the achromatic optics, and the origins of the concept of cell biology developed and formulated by Matthias Jacob Schleiden for plants and by Theodor Schwann for animals. In this review, by sketching the history of NMJ research from its start in the 1840ies we have aimed to demonstrate the power of this important synapse for the progress of malleable scientific concepts as well as some major technological advancements that were necessary to conclude long debates on uncertain scientific aspects. In doing so, we have concentrated on aspects of early anatomy, function, and principal molecular components of vertebrate NMJ. In particular, this review delves into the literature on sympathetic innervation of NMJ, since this was intensely studied and debated for the first third of the twentieth century and then almost completely abandoned until recently. Other topics such as NMJ development, glial function, and neuromuscular transmission disorders have been lately covered in excellent review articles, and have therefore been handled in a cursory manner in this review with due citation. We apologize for not mentioning many authors or works that could have been also included but were not due to space limitations.

## 2. Phase I: Discovery of the NMJ

In 1839, Theodor Schwann founded the “cell theory” which states “[…] that a common developmental principle underlies all individual elementary particles of all organisms” and that organismal growth is solely based on the generation of new cells as well as the extension of existing cells [3]. Although the term “cell” had been in use by other researchers before, it had been ill defined. Some saw in it the cell nucleus, others the cell body or yet further structures. In this same period, in which the concept of cell biology was just beginning to evolve, Louis Michel François Doyère and Jean Louis Armand de Quatrefages de Bréau depicted perhaps the first NMJs in the literature [4,5]. They show endplates of the tardigrade *Milnesium tardigradum* and the snail *Eolidina paradoxum* and are drawn in continuity with the muscle fiber (see Figure 1A,B).

Explicitly, de Quatrefages speaks of a “reciprocal penetration” and a “veritable fusion of nervous and muscle tissues” [5]. Many researchers after Doyère and de Quatrefages, including Willy Kühne, L. Bremer, Jan Boeke, and Louis-Antoine Ranvier, also talked of “hypolemmal” nerve endings to express their opinion that the nerve fiber terminated below the sarcolemma. This view was in contrast to the “cell theory” by Schwann and the “neurone doctrine” by Ramón y Cajal, but in line with the “reticular theory” heralded by Joseph von Gerlach and Camillo Golgi [7], which proposed the nervous system as a network of neurons one passing into the other in an anastomosing manner. Although preparation techniques and microscopic anatomy improved a lot in these years, the existence of two contradicting theories casted shadows on the Nobel prize ceremony in 1906. There Cajal and Golgi jointly received the award for Medicine and Golgi was utterly arguing in his speech against the “neurone doctrine”. Notably, the precise arrangement of nerve and muscle cell at the NMJ remained open for many years to come. Those favoring the “epilemmal” conformation, i.e., where nerve and muscle would be separated from each other, were rather few, while many advocated for an “hypolemmal” situation whereby the nerve would enter the sarcolemma. This point was important, because it strongly influenced the way of how signal propagation from nerve to muscle was conceived (see chapter III). Ultimately, upon development of electron microscopy, the argument was terminated for the NMJ in the 1950ies in favor of the “neurone doctrine” with the first descriptions of synaptic clefts [8,9] and a three-dimensional reconstruction of the NMJ ultrastructure [10] (see Figure 2 for an overview of phases).

Apart from this intense debate, some further fundamental anatomical discoveries were made on the motor endplate in the period between 1840 and 1900. For example, Willy Krause, an important exponent of the “epilemmal” conformation hypothesis, coined the term “motorische Endplatte” [11] and found in cat muscle that each fiber had a maximal length of about four centimeters and one single endplate located roughly in the middle of the fiber [11]. Although in general, the latter is a finding that seems to hold true even for large animals such as elephant and buffalo [12], it does not apply to all muscle types. For example, Zenker and colleagues found in rat that sternomastoid muscle indeed exhibited a single NMJ band in the middle of the muscle, whereas gracilis anterior and latissimus dorsi muscles showed partially multiple innervation per fiber and myo-myonal junctions [13]. Charles Marie Benjamin Rouget reported on the granulated fine structure of the synapse [14,15], anticipating the presence of synaptic vesicles. At the NMJ, Ranvier distinguished “fundamental nuclei” of the muscle fiber from “arborization nuclei” [16], the latter likely being nuclei of terminal Schwann cells. Schwann cells and their myelin sheaths were described by Schwann and Ranvier [3,16]. Kühne, using gold impregnation methods based on those developed by Golgi [17], beautifully depicted hundreds of NMJs from many different vertebrates ranging from amphibians over reptiles and birds to mammals at high resolution [6] (see Figure 1C for some examples). He described the end-ramifications of NMJs as “antlers” and categorized them according to their ramification modes. Furthermore, he observed the “Borstensaum”, bristle-like borders on the endplate ramifications of vertebrates that were likely representing postsynaptic junctional folds [6]. Finally, Bremer gave a first account on a dual innervation of NMJs by myelinated and thin axons [18]. As outlined in the following chapter, this instigated another long dispute, which is actually still open today, on the nature of the motor endplate.

## 3. Phase II: Small vs. Myelinated Fibers, Plastic Tone vs. Voluntary Movement

In 1894, Sir Charles Scott Sherrington showed that nerves supplying skeletal muscles contain axons of different diameters, myelinated and non-myelinated, dorsal afferent and ventral efferent, sensory, somatic, and sympathetic [19]. He also found that the largest myelinated axons were of motor type, while another group of myelinated axons were connected to muscle spindles [19]. The latter had been described and termed as such some 20 years earlier by Kühne [20]. Thus, by the end of the nineteenth century, the major neuronal components and, particularly, the presence of somatic and sympathetic innervation of skeletal muscle were established. Also, based on observations from Robert Remak in 1837 [21], it was known by Sherrington that motor axons were large-sized and myelinated; conversely, sympathetic axons appeared as pale and non-myelinated [19]. However, while motor neurons were easily found to induce voluntary contraction of skeletal muscle, the function of sympathetic neurons in muscle tissue remained unclear. Apart from the universally accepted function of sympathetic innervation on vasomotor activity [22,23,24,25], early findings of Bremer indicated also an intimate interaction of large myelinated and small non-myelinated nerve fibers in the region of motor endplates [18]. In 1902, Aldo Perroncito, a coworker of Golgi, suggested that the small fibers at the NMJs be of sympathetic origin [26]. This was further supported by a series of morphological and some experimental data, which were chiefly contributed by Boeke [27,28,29], Erik Agduhr [30], and Marion Hines [31]. Boeke termed such presumably sympathetic inputs to endplates “accessory fibers” [29] (“af” in Figure 3A).

In the first three decades of the 20th century, the discussion about the existence of a dual innervation of skeletal muscle fibers and of NMJs by somatic myelinated motor neurons and sympathetic non-myelinated neurons, was very intense and controversial. While the founder of the term “autonomic nervous system” and one of the most influential researchers in this field, John Newport Langley, admitted that the morphological evidence was indeed in favor of the existence of a dual innervation, he was rather doubtful with respect to the experimental findings [23] and the puzzling unanswered question about the potential role of such a dual innervation. Increasingly, this casted shadows on the dual innervation hypothesis. Thus, in the 1920–30ies three major schools of thought evolved on this topic: First, there was the group of those researchers who fully supported the concept of dual innervation of NMJs. Second, some others neglected the existence of dual innervation of the same muscle fiber and motor endplate, but suggested that distinct groups of muscle fibers would be innervated by either motor or sympathetic neurons and form either plate-like or grape-like endings [32,33]. These different fiber groups were thought to be responsible for distinct types of contractility, i.e., voluntary and reflex activities for the muscle fibers with plate-like endings and plastic tone activities for those with grape-like endings. A third school of authors, heralded by Joseph B. Hinsey [34], doubted the existence of sympathetic endings on muscle fibers altogether and suggested that the structures observed by Boeke, Agduhr, and others were artefacts [35,36,37].

A principal issue with the studies of Boeke and others regarding the sympathetic co-innervation of NMJs was the lack of molecular markers. At that time, anatomical studies were carried out using impregnation with silver, gold, or other dyes. Although these were apparently enriched in nervous tissue, they would not unequivocally discriminate between different types of neurons and, partially, were considered to label also connective tissue [34]. Thus, the only way to further distinguish between different neuron identities was by analysis of axon diameter and myelination, which again did not indicate cell-type identity unless combined with functional studies such as specific denervation experiments. However, also here, findings were equivocal with respect to a sympathetic co-innervation of NMJs [37,38] and, consequently, the interest of the research community in the field of sympathetic innervation of skeletal muscle and the NMJ rapidly declined in the mid 1930ies. Since then, for the next eight decades, the school favoring the absence of a direct function of sympathetic innervation in the skeletal muscle tissue has dominated the field. In the context of this mindset, the interpretation by Chan-Palay et al. in 1982 of their immunohistochemical data is maybe not too surprising. Indeed, they investigated the presence of different synthesizing enzymes at the NMJ and found a consistent enrichment of the sympathetic neuron marker, tyrosine hydroxylase (TH), at the NMJs of different species, including human. However, instead of considering the possibility of an accessory innervation of NMJs by sympathetic neurons, the presence of TH at NMJs was considered to be an additional biochemical capacity of lower motor neurons [39,40].

Between the 1930ies and the present time, one of the very few extensive papers on sympathetic innervation of skeletal muscle showed the presence of direct sympathetic innervation on skeletal muscle fibers and muscle spindles [41] on the basis of data using the Falck-Hillarp fluorescence method to reveal biogenic monoamines [42,43,44] in combination with ultrastructural features of sympathetic neurons. While that work did not address sympathetic innervation at NMJs, two studies in 2013 and 2016 reported TH immune-positive axons approaching the NMJs of adult mouse hindleg muscles [45,46]. TH-positive axons were mostly ending in plaque-like structures opposite to the postsynaptic, pretzel-like arborizations of nicotinic acetylcholine receptors (nAChRs). Notably, though, instead of perfectly matching the distribution of nAChR staining as it would be the case for motor nerve terminals, the TH-positive plaques were often complementary to the NMJ staining signals [45,46]. The distribution of sympathetic neurons was confirmed by genetic reporter mice and additional immunostaining of another sympathetic marker protein, neuropeptide Y [46]. Functional coupling between sympathetic innervation and NMJs was shown using a combination of direct electrical stimulation of lumbar sympathetic ganglia and simultaneous measurement of NMJ postsynaptic activities. This revealed immediate, sympathetic-stimulus dependent postsynaptic activation of β-adrenergic receptors and cAMP signaling as well as import of the transcriptional co-activator PGC-1α [46]. Fittingly, local chemical ablation of sympathetic innervation for two weeks by intramuscular injection of the sympathicotoxic drug, 6-hydroxy dopamine, had gross effects on NMJ morphology and compound muscle action potentials, and induced massive muscle atrophy [46]. These alterations could be largely rescued by simultaneous treatment with the sympathicomimetic drug, clenbuterol [46], suggesting the presence of sympathetic innervation of NMJs in mouse hindlimb muscles with functions in NMJ maintenance, electrophysiological properties, and muscle trophic status. Recently, this hypothesis was confirmed and expanded to functional roles of sympathetic innervation on motor neurons and to better understand the molecular postsynaptic signaling activated upon norepinephrine release [47]. In particular, surgical sympathectomy was found to alter acute neuromuscular functionalities, including nerve-evoked contractile force, amplitudes of miniature endplate potentials (MEPP) and endplate potentials (EPP), as well as quantal content [47]. Accordingly, application of sympathicomimetics, salbutamol or clenbuterol, increased these parameters, and they were found to depend on enhanced Ca^2+^ influx through transient receptor potential channels V1 and P/Q-type Ca^2+^ channels [48]. Fitting to the observed effects on neuromuscular transmission of sympathectomy, the axons of motor neurons of treated animals presented with a disorganized cytoskeleton, myelination defects, and smaller axon diameters, as well as a fiber-type specific reduction of muscle cross-sectional area [47]. With respect to molecular pathways involved at the postsynaptic side, mRNA-profiling revealed that genes associated with muscle denervation such as the γ-subunit of nAChR, and the myogenic transcription factors, myogenin and MyoD, were significantly up-regulated upon acute surgical sympathectomy, whereas genes involved in vesicle docking and fusion, circadian rhythm, Notch signaling, and energy homeostasis were diminished [47]. Finally, surgical sympathectomy was found to induce a Hdac4-myogenin-MuRF1-miRNA-206 expression axis. Based on previously described roles of MuRF1 on muscle atrophy [49,50,51] as well as on nAChR recycling and degradation [52,53], this was suggested to explain the observed massive muscle atrophy and loss of plasmamembrane-bound nAChR [47]. Concerning the postsynaptic signaling cascade potentially triggered by sympathetic activation of β2-adrenergic receptors, the involvement of a cAMP-protein kinase A axis as well as of G_αi2_ were proposed [47]. Another two recent studies investigated postnatal development of sympathetic innervation of NMJs and its potential role on NMJ maintenance upon aging. One of these showed the presence of TH-positive neurons already at birth for different muscles throughout the body, including diaphragm and hindleg muscles [54]. While TH and neuropeptide Y signals at NMJs strongly increased during the first postnatal weeks in extensor digitorum longus muscles, β2-adrenergic receptor density at endplates was high at birth and then slightly decreased in mature muscles [54]. The second study reported that genetically engineered brain-specific Sirt1-overexpressing (BRASTO) mice leading to enhanced sympathetic outflow showed increased enrichment of TH and β2-adrenergic receptors at NMJs as well as reduced fragmentation of motor endplates in sternomastoid muscles of aged BRASTO mice when compared to wildtype animals [55]. Conversely, down-regulation aggravated morphological defects, particularly, with respect to terminal Schwann cells [55]. A brief overview about the current findings with respect to the function of the sympathetic innervation at NMJs can be found in Figure 3B.

In summary, a series of recent anatomical, morphological, and experimental studies has started to re-establish the validity of the hypothesis of sympathetic co-innervation of NMJs. Future work is needed to further consolidate these findings and to deepen the insights into the functional relevance of dual innervation in health, disease, and aging. It will be important to better characterize acute and long-term effects of sympathetic innervation, to determine the target cells, and mechanisms of action.

## 4. Phase III: ACh and NE—Neurotransmitters at the NMJ

The second half of the nineteenth century had revealed several principal anatomic features of the NMJ, but the mode of interaction between nerve and muscle had remained elusive. In particular, it was still unclear, if motor neurons would terminate in a “hypolemmal” or “epilemmal” manner. In the hypolemmal configuration, even a direct electrical conduction from nerve was envisioned. However, studies using the South American arrow poison curare, which gained strong attention through the meticulous work and description of Claude Bernard [56], were rather advocating for a chemical transmission. Indeed, Edmé Félix Alfred Vulpian showed in the 1860ies that upon application of curare, (i) nerve excitation did not pass on to the muscle, (ii) curarized muscle remained directly excitable, and (iii) curarized nerve could transmit its excitation to a non-curarized muscle [57,58]. In 1877, these and other observations as well as further theoretical considerations led Emil Heinrich du Bois-Reymond to critically examine the direct wave-like propagation of electrical stimuli from nerve to muscle and to reason about the possibility of a chemical transmission [59]. At the end of the nineteenth century, Sherrington elaborated on such ideas to formulate the concept and term of “synapse” [60,61]. Today, chemical transmission from motor nerve to muscle is a common place. Back then, though, the identification of the signaling molecules involved was rather indirect and for the NMJ the issue was finally solved only in the mid twentieth century.

The initial major steps forward came from work on the autonomic nervous system: Indeed, an important notion here was that organs supplied by both sympathetic and parasympathetic branches of the autonomic nervous system would typically respond to stimulation of one of the two in an opposite manner. For example, heart beat would increase and decrease upon excitation of the sympathetic and parasympathetic nerves, respectively. As for the sympathetic part, adrenal extracts from dogs were found to be highly active on blood pressure, heart beat, respiration, and skeletal muscle contraction [62]. Around the year 1900, adrenalin or epinephrine, the active principle in these adrenal extracts, was first isolated by John J. Abel in an impure form [63] and then, simultaneously, in pure version by the Samurai and chemist Jokichi Takamine [64] and Thomas Bell Aldrich [65]. However, adrenalin’s function did not perfectly recapitulate stimulation of sympathetic nerves, suggesting that it was similar but not identical to the sympathetic neurotransmitter. In 1946, the real neurotransmitter was found by Ulf Svante von Euler-Chelpin and characterized as “sympathin”, which we now call noradrenalin or norepinephrine [66,67,68].

The active compound released by parasympathetic endings, acetylcholine (ACh), had been synthesized from brain extracts by Adolf Baeyer already in 1867 [69,70], but its physiological role on blood pressure was only discovered in 1906 by Reid Hunt and René de Taveau [71]. In 1921, Otto Loewi stimulated either parasympathetic Vagus nerves or sympathetic nerves of frog hearts, took the media from these stimulation periods, and called them “Vagusstoff” and “Acceleranzstoff”, respectively. Their application to untreated hearts showed that they could mimic the stimulation of the corresponding nerves [72] and thus convincingly demonstrated that there is chemical transmission from nerve to heart. Although he also speculated that “Vagusstoff” could be ACh, this first remained unclear. With respect to NMJ, Walter Rudolf Hess found in 1923 that application of ACh induced skeletal muscle contraction, which was comparable in strength to that upon motor nerve stimulation [73]. Since this occurred in an atropine-independent manner, it did not relate to parasympathetic action [73]. In 1934, Sir Henry Hallett Dale and Wilhelm Siegmund Feldberg showed that ACh was released upon motor nerve stimulation [74] and together with Marthe Louise Vogt, they demonstrated that ACh release correlated with muscle contraction and that this function was inhibited by curare [75]. Simultaneous to the progress with respect to ACh activity, its hydrolytic decay by acetylcholine esterase [76] at the NMJ [77] was characterized. Thus, around one hundred years after the first observation of NMJs, the neurotransmitter chemistry was basically clarified. The molecular identification of the nAChR and many of its major interacting proteins occurred decades later and strongly relied on electric fish as “NMJ-enriched” model systems [78,79] (see chapter V). Conversely, the electrophysiological processes underlying the neuromuscular transmission were largely worked out on frog and cat NMJs and although they started even earlier they were concluded with a significant time shift compared to the studies on transmitters. Presumably, this was mainly due to the persisting uncertainty about the relationship between nerve, muscle, and the NMJ. Indeed, a leading electrophysiologist of these times, John Carew Eccles, still wrote in 1939: “At the neuro-muscular junction the motor nerve fiber penetrates the sarcolemma and forms an ending of variable complexity in a specialized region of the muscle fiber called the motor endplate.” And: “The state of the true surface membranes of the muscle fiber and motor endplate is unknown, being, on account of its ultramicroscopic character, more a physiological than a histological problem” [80]. Consequently, the precise nature of the information flow from nerve to muscle including presynaptic release of ACh over its binding to postsynaptic receptors and their triggering of a muscle action potential, was not really understood until the 1970ies. Most of the key steps occurring during neuromuscular transmission were unraveled by a small group of people, with Bernard Katz being their main exponent.

## 5. Phase IV: MEPP, EPP, and AP—Conversion of Chemical Signals into Electrical Activity

Initial observations and interpretation of muscle action potentials were made in the 1840ies by Carlo Matteucci [81] and du Bois-Reymond [82]. Conduction speed of action potentials along peripheral nerves was first and correctly measured to be around 27 m/sec in cold frog sciatic nerve by Hermann von Helmholtz in 1852 [83]. Nerve conduction principles were studied further in an intensified manner in the early 20th century by Sherrington and Edgar Douglas Adrian. For example, while Sherrington contributed many principal aspects about the systemic and integrative functions of nerve conduction [61], Adrian described the summation of stimuli, the refractory period of stimulation, and the all-or-none principle of neuron firing [84,85]. These increasingly fine electrophysiological measurements requested increasingly sensitive technologies. As a side effect of the first World War research, electrophysiological instruments were dramatically improved in sensitivity in the 1920ies by virtue of novel vacuum [86] and three-stage amplifiers [87] as well as cathode ray oscillographs [88]. They allowed to record from single neurons and to differentiate between individual neural activities in a nerve.

As described in the previous chapter III, around 1920 it was still completely open how the motor nerve would propagate its stimulatory signal to the muscle, and the potential involvement of ACh was just started to be discussed. This began to change in the late 1930ies by the first observations of EPPs in frog [89] and cat [80] and the understanding of an involvement of ACh in this process [90,91,92]. In 1952, the recording of MEPPs with an intracellular measurement electrode by Fatt and Katz constituted an important step towards the “quantal hypothesis” [93], which was formulated by del Castillo and Katz in 1954 [94]. It was clear now that synaptic transmission is mediated by the transmitter acetylcholine, which is released very locally from the presynaptic motor nerve and interacts with receptors on the postsynaptic membrane. With the advent of ultrastructure research and the observation of the numerous synaptic vesicles in the presynaptic terminal [8,9], the “vesicle hypothesis” was brought forward. This stated that “ACh may be contained in the vesicles and released in an all-or-none manner when the vesicle collides with certain spots of the terminal membrane” [95]. By combining cell fractionation by differential and density-gradient centrifugation with electron microscopy, Whittaker and colleagues prepared so-called “synaptosome” preparations, which consisted of nerve endings containing most of the particle-bound “vesicular” ACh [96]. The presence of ACh within synaptic vesicles from brain was confirmed some ten years later by Whittaker and colleagues [97,98]. The enrichment of synaptic vesicles in “active zones” was described in the frog NMJ by Couteaux and Pécot-Dechavassine [99] and was beautifully depicted four years later using freeze-fracture preparations by Heuser, Reese, and Landis [100]. From their electron microscopic studies using resting and stimulated muscles, Heuser and Reese also extracted compelling morphological evidence for the existence of synaptic vesicle recycling [101], which proved to be perfectly correct [102]. In parallel to the groundbreaking work on the molecular synaptic vesicle machinery (reviewed in [103]), Harlow and McMahan used ultrastructural tomography of frog NMJs [104,105] to propose a mapping of morphologically identified active zone material to molecular identities [106]. On the postsynaptic side, Takeuchi showed ACh-dependent endplate currents involving sodium and potassium [107]. These were further carefully analyzed by Katz and Miledi in 1970, leading to the first observations of elementary events of nAChRs opening and closing upon ACh binding [108]. Iontophoretic mapping revealed an extremely high enrichment of nAChR at the NMJ, since sensitivity to local application of ACh was high at the postsynaptic surface but very low just next to it [109,110]. Significant technological advancements by Sakmann and Neher led to the development of the patch-clamp method [111]. This permitted to study the electrophysiological characteristics of nAChR in unprecedented detail and gave the first comprehensive account of single-channel measurements [112]. With these studies, NMJ research had definitely reached the molecular level, whose hallmarks are outlined in the next chapters.

## 6. Phase V: Nerve—Muscle Signaling—Feed Forward

While patch-clamp technology yielded precise information on the electrophysiological characteristics in the 1970s, initial speculations about the existence of a cholinergic receptor on muscle were drawn much earlier from pharmacological approaches. In 1905, Langley applied nicotine to anaesthetized fowl with sciatic nerve sectioned on one side and observed tonic contracture of leg muscles on both innervated and denervated legs unless antagonized by curare [113]. Notably, the same was true for isolated gastrocnemius muscles, demonstrating the presence of a “receptive substance” on the muscle that (i) can mediate muscle contraction, (ii) is activated by nicotine, and (iii) is inhibited by curare. In 1914, Dale noted distinct “muscarinic” and “nicotinic” actions of acetylcholine, which were chiefly found for target organs of the parasympathetic nervous system and on skeletal muscles, respectively [114]. This suggested the existence of at least two cholinergic receptor classes with different pharmacology, function, and localization. We know now that the first class, muscarinic receptors (mAChR), make up a family of five different metabotropic G protein-coupled receptors (M1-M5), which are either inhibited by scopolamine and/or atropine, and regulate heart rate, smooth muscle contraction, glandular secretion as well as many fundamental central nervous functions [115]. At the NMJ, muscarinic receptors are involved as presynaptic autoreceptors in the context of early postnatal synaptic pruning were they cooperate with purinergic and Tropomyosin receptor kinase B receptors to reinforce strong synaptic contacts and suppress weak ones in autocrine feedback loops [116,117]. The second group, nicotinic acetylcholine receptors (nAChR) are pentameric ion channels and are grossly grouped into neural/epithelial and muscle nicotinic receptors. In general, they are inhibited by curare and are composed of five subunits.

The isolation, purification, and biochemical characterization of nAChR could be realized applying newly developed methods of cell fractionation by differential centrifugation [118,119,120], electrophoresis methods, the availability of electron microscopes, the synthesis of detergents such as Triton X-100, and denaturing agents such as sodium dodecyl sulfate. Most important, though, was the help of nature providing toxins that bind with high specificity to nAChR, such as the elapid snake venom α-bungarotoxin, as an exquisite binding agent of nAChR [121,122,123]. In addition, the electroplax from electric fish such as *Electrophorus electricus* and *Torpedo* species were recognized as an excellent source of cholinergic synapse components [78,124]. The close relationship of electrocytes of the electric organ to muscle had been described already in the 1890ies and the assumption that their signal transmission was of cholinergic nature was confirmed in the 1940ies, for example by Feldberg and Fessard [125]. In postsynaptic membranes of Torpedo electric organs, nAChR make up 40–50% of total protein, thus, rendering this a convenient starting material for purifying synaptic constituents. Further fractionation procedures yielded postsynaptic membrane fragment preparations (“microsacs”) that were highly enriched in nAChR [126]. These could be solubilized and further purified e.g., by means of affinity chromatography with the receptor retaining its functional properties (inter alia [121]). The clarification of the subunit composition of the isolated receptor complex, however, turned out to be difficult because gel electrophoresis results appeared very variable due to preparation artefacts, uncontrolled proteolysis, or other unknown differences. In 1974, the groups of Karlin [127] and Raftery [128] presented data that indicated four subunits, in a putative ratio of 2:1:1:1. Finally, the protein sequences of all four nAChR subunits were determined after another six years and these conclusively showed the nAChR as a pentameric complex consisting of 2 × 40 kDa, 1 × 50 kDa, 1 × 60k Da, 1 × 65 kDa [114]. Based on these findings, cloning of the corresponding cDNA sequences was enabled [129,130,131,132,133]. The hetero-pentameric, α_2_βγδ composition of nAChR from *Torpedo* electroplax as well as mammalian muscle was further supported by cryo-electron microscopy and later by crystallographic methods [134,135,136]. A novel, mammalian-specific ε subunit was cloned in 1985 [133] and found to be characteristic of innervated adult muscle [137,138] with strictly subsynaptic expression [139,140]. Many important components of the NMJ, including rapsyn [141,142], agrin [143], MuSK [144], and LRP4 [145] were identified in the following years. The rich field of research that has studied many of the interactions and downstream effectors of these proteins as well as their involvement in NMJ development, maintenance, and aging has been recently described in excellent reviews [146,147] and is therefore not further discussed here.

## 7. Phase VI: Postsynaptic Apparatus—nAChR Turnover

In parallel to the discovery of the nAChR and its binding partners, the field of molecular cell biology became increasingly strong. Given the relatively easy access to NMJs, the development of radioactive, fluorescent, peroxidase, or biotin-labeled α-bungarotoxin species [148,149,150,151] and the concomitant advances in electron and fluorescence microscopy, the study of the postsynaptic apparatus at the NMJ became a driving component in neuroscience research for the better understanding of molecular machineries at synapses. As such, work on nAChR revealed major general features of neurotransmitter receptors, including receptor clustering, receptor localization and regulation of receptor turnover. With respect to the latter, muscle denervation was soon found to induce massive production of new nAChR, which were largely located in extrajunctional regions (reviewed in [152]). Furthermore, use of radiolabeled α-bungarotoxin showed that nAChR lifetime was strongly dependent on muscle innervation status [123,153]. While the half-life of nAChR was several days in normal innervated muscle, it went down to only approximately 24 h upon denervation. This finding was later reproduced by several other studies using further, refined radiolabeling [154,155,156,157,158,159,160,161,162] or live-animal imaging methods with fluorescently labeled α-bungarotoxin in wildtype [163] and in transgenic nAChRγ-GFP mice [164]. The observations of nAChR endocytosis using electron microscopic analysis of electron-dense precipitates mediated by peroxidase-coupled α-bungarotoxin [165,166] and of a nAChR recycling pool by a smart α-bungarotoxin-biotin streptavidin-dye live imaging approach [167], paved the way for a series of further studies on the pathways leading to nAChR recycling and degradation. These used live imaging in combination with sequential application of differentially labeled α-bungarotoxins and often with heterologous expression of genetically encoded molecular biosensors. As reviewed recently [168,169], these studies identified a complex on nAChR recycling vesicles composed of rapsyn, myosin Va, and protein kinase A [170,171,172,173]. Apparently, this complex got activated during the early postnatal period in mouse [174]. This period is also characterized by a switch of nAChR subunit composition [175], synapse elimination [176,177,178], and an increase in structural complexity [179,180], which is largely mediated by podosome formation [181]. Furthermore, recycling of nAChR was found to be under control of protein kinase C [173] and Ca^2+^/calmodulin-dependent kinase II [182]. Instead, autophagy was reported to underlie the process of nAChR degradation [183,184]. This involved a complex of the E3-ubiquitin ligase MuRF1, the autophagy adaptor p62, the autophagosome-formation factor Bif-1, and the endosomal regulator Rab5 [52,53,185]. Notably, a recent work has shown that nAChR turnover and MuRF1 expression are strongly altered upon sympathectomy [47], suggesting that these processes are under strict sympathetic control. It has been debated, if the regulation of nAChR turnover by sympathetic innervation could explain the clear beneficial effects of sympathicomimetic treatments of patients suffering from several forms of congenital myasthenic syndromes (CMS) [169,186,187], a group of inherited neuromuscular disorders. However, it is early days to make final conclusions here. In general, the fields of research on CMS and on the molecular machinery of the NMJ have stimulated each other quite a lot. Based on the current knowledge, roughly 30 genes can be affected in CMS and these were recently grouped into six major categories, i.e., (i) presynaptic, (ii) synaptic space, (iii) postsynaptic, (iv) defects in NMJ development and maintenance, (v) defects in glycosylation, and (vi) other syndromes [188]. Given that 18 of the genes involved in CMS were identified in the short period between 2012 and 2018, it is to be expected that an increasingly precise understanding of mechanisms at the neuromuscular apparatus in physiology and disease will follow from this synergy in the coming years.

## 8. Phase VII: The Future of NMJ Research

It is difficult to predict the future directions of NMJ research, but owing to the apparent healthcare needs in Western societies and/or recent scientific advancements, three fields appear to be rather promising to these authors. However, as the fields of mouse genetics, optical tissue clearing, and imaging have made such rapid progress in the past few decades, it appears to be a good time to intensify studies of more systemic character. To begin with, the findings of NMJ fragmentation upon aging [189], the reduction of aging-related fragmentation by caloric restriction [190], and advancements in the understanding of the cellular [191] and molecular [184,192,193,194] mechanisms underlying this fragmentation, there has been a growing interest to study the relationship between NMJ and sarcopenia. This topic has been recently covered in several reviews [169,195,196,197]. Maybe still less explored, but potentially relevant for the understanding of clinical features in certain motor neuron diseases, is the concept of retrograde signaling from muscle to nerve. Indeed, different muscle-derived proteins were found to be crucial for proper presynaptic development and NMJ positioning, at least in mouse. These proteins could be grouped in transmembrane receptors (MuSK, LRP4, β1 integrin) [198,199,200,201], ECM components (collagens IV and XIII, laminin β2) [202,203,204,205], and secretory signaling factors (FGF 7 and 10 and 22, Slit2) [202,206]. If similar mechanisms are critical in human beings, they might help to explain why sporadic forms of the classical motor neuron disease, amyotrophic lateral sclerosis (ALS), often show distal to proximal spreading of axonal conduction changes [207]. Whether muscle-driven NMJ degeneration followed by retrograde aberrant signaling to the motor neuron can be involved in such a scenario of “dying back” is still elusive. However, first studies using mice with conditional muscle-specific expression of the ALS target gene, superoxide dismutase, showed fragmentation of NMJs [208] and motor neuron loss [209]. Finally, it will be interesting to see what can be unraveled about systemic interactions at the NMJ between the different types of innervation, as well as contributions from terminal Schwann cells [178], kranocytes [210], muscle, blood vessels etc. The skeletal muscle tissue has often been considered to be a simple system that is just good for robust responses to motor nerve impulses. However, such a reductionist view does clearly not reflect the seamless adaptation of this system to the multiple types of strain and fine control by the different participating components that render skeletal muscle our mainstay through decades of life. We are looking forward to many new discoveries on the motor endplate, the centennial model synapse and its embedding muscle tissue.

## Figures and Tables

**Figure 1 cells-08-00387-f001:**
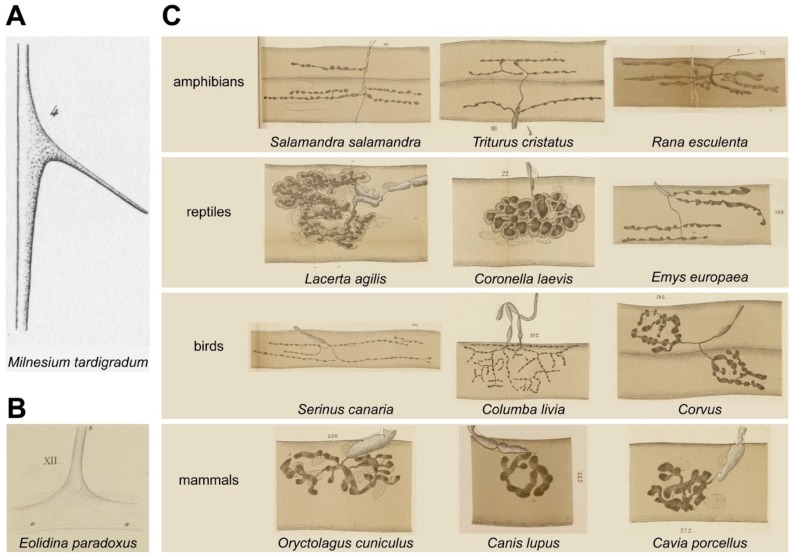
Historical anatomic drawings of NMJs. A-B: Presumably the first drawings of NMJs from the early 1840ies. Figures show reprints of NMJs from the tardigrade, *Milnesium tardigradum* (**A**) and the snail, *Eolidina paradoxus* (**B**). Reproduced from [4,5], original sources: www.biodiversitylibrary.org/item/47974#page/7/mode/1up and www.biodiversitylibrary.org/item/47973#page/9/mode/1up. (**C**). Selection from the more than 300 NMJs depicted by Willy Kühne in 1886 shows the variety of vertebrate NMJ morphologies. Reproduced from [6], original source: https://catalog.lib.uchicago.edu/vufind/Record/11720295.

**Figure 2 cells-08-00387-f002:**
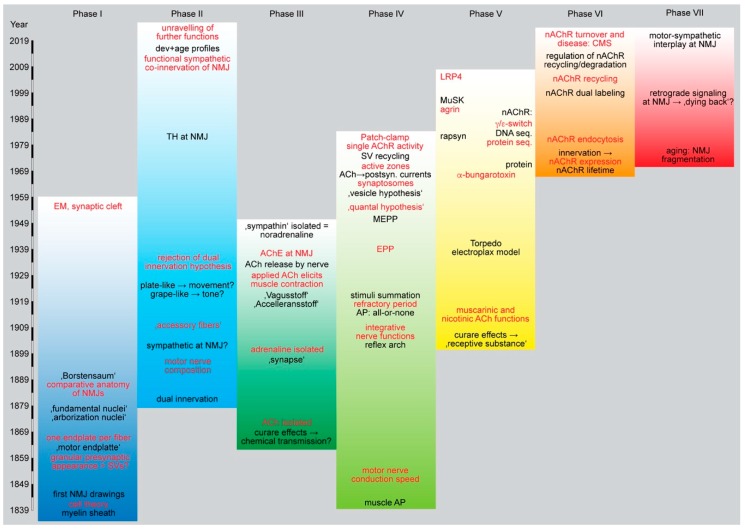
Schematic overview of the different phases of NMJ studies. This scheme highlights major aspects of research on NMJ.

**Figure 3 cells-08-00387-f003:**
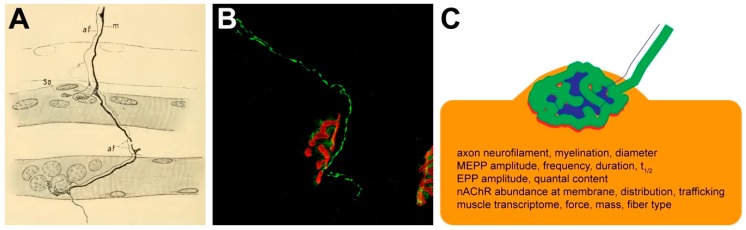
Past and current of sympathetic innervation of NMJ. (**A**) One of the first observations of sympathetic co-innervation. Reproduction from [29], original source: www.biodiversitylibrary.org/item/43338#page/7/mode/1up. Abbreviations: af, accessory fiber; m, motor fiber; so, sole plate. (**B**) Fluorescence micrograph showing sympathetic innervation (green, TH immunostaining) of NMJ (red, alpha-bungarotoxin staining) in mouse tibialis anterior muscle. Reproduced from [54]. Image size, 180 µm. (**C**) Scheme depicting the putative complementary localization of sympathetic (blue) and motor neuron endings (green) at NMJ. While the latter is perfectly fitting the shape of the postsynaptic nAChR distribution (red), the sympathetic neuron signals are rather found in between. The differential thicknesses of blue and green axons indicate the absence and presence of a myelin sheath, respectively. In addition, the diameter of sympathetic axons is normally also smaller than that of motor axons. The writings indicate major functional roles of sympathetic innervation at NMJ as presently known.

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
