# Peer review of "Motor Endplate—Anatomical, Functional, and Molecular Concepts in the Historical Perspective"

_cells, 2019, doi:10.3390/cells8050387_

Round 1

Reviewer 1 Report

Motor Endplate - the Centennial Model Synapse:

Principal Concepts and Technical Advancements

By Rüdiger Rudolf, Muzamil Majid Khan and Veit Witzemann.

Because of   the relevance of the NMJ as an attainable model of synapse (several basic   mechanisms of synaptic organization and function have been first described in   the NMJ) many reviews are wrote concerning different developmental,   molecular, physiological, structural, and pathological aspects. However, the present   work make a significant contribution to the field with the accurate revision   of the historical evolution of the knowledge from 1840.

The review   is comprehensively wrote and the authors apologize for not mentioning many works.   This is easy to understand because the wide and plentiful period covered. However,   at least some comment on several important topics of the NMJ function could   improve the review. For instance but not only:  i) NMJ development (axonal competition and   synapse elimination have been also observed in all neural tissues), ii)   Neurotrophins and retrograde control of the neurotransmission linked to postsynaptic   cell activity, iii) Presynaptic autoreceptors (muscarínic and purinergic),   iv) The molecular machinery of the synaptic vesicle exocytosis v) The involvement   of the teloglial cells in the synaptic function.

The work is   scientifically sound and the English used seems correct to this reviewer.

Author Response

We are grateful for the positive overall judgement of the manuscript and the helpful and constructive suggestions. In the following, please find the modifications to the text to accommodate most of your requests:

At lines 380 ff. we added a remark on the autocrine functions of mAChR and TrkB, and purinergic receptors in the context of NMJ development.

At lines 353 ff. there is a short account on the molecular machinery of synaptic vesicle exocytosis.

At lines 485 ff. terminal Schwann cells were included in the general consideration of NMJ functionality; however these were also mentioned before at other places (e.g. on line 443).

We have refrained from extending this review more towards NMJ development, because there is a plethora of recent reviews on this topic. We have, however, cited many of these reviews and have now also added a specific remark at lines 50 ff. to explain our choice.

Reviewer 2 Report

There is much to commend this review article, especially from its historical account (which filled some gaps in my own knowledge),  its critical evaluation of the sympathetic innervation of NMJs, and the account of the molecular biology of ACh receptors. These are topics that I know are of particular interest to the authors and to which they have made important original research contributions. But therein also lies the weakness of the review: it is unbalanced and, for a general review implied by the title, there is too much emphasis on these aspects of NMJ biology at the expense of others. The introduction and historical review (Phase I) are extremely good. For me, the problems start with Phase II, describing the innervation pattern of muscle at NMJs. This rapidly becomes diverted into a very detailed case for sympathetic co-innervation of NMJs, to the point that some readers may be forgiven for thinking that this is the main source of the nerve supply and not that by somatic motor neurones! The next sections (Phases III and IV), on neurotransmitter release and their postsynaptic effects, provide a useful historical account of chemical synaptic transmission. It is a pity, however, that - perhaps due to the focus on vertebrate NMJ - the authors make no reference to the fantastic recent technical developments that have enabled visualisation of transmitter release and action at neuromuscular junction, provided by the targeted expression of GCaMP variants to postsynaptic membranes at larval Drosophila NMJs (eg the work of the Littleton and Isacoff groups at MIT and Berkeley respectively). This technology has, literally, added a new dimension to investigation of mechanisms of transmitter release since it allows spatial as well as temporal investigation of neuromuscular synaptic function; so I am surprised that no reference to this work has been included, especially since the title of the article refers to "technical advancements". These two sections are also not as detailed in their analysis of the role of diverse presynaptic proteins as provided for postsynaptic proteins and especially nicotinic AChR at NMJs in the final two analytical sections (Phase V and VI). In terms of future directions, it's rather curious that the focus suddenly turns to aging/ sarcopenia and to ALS, when the disease targets based on the content of the foregoing sections would seem to militate for a greater focus on the various forms of myasthenia and myasthenic syndromes: not least given that salbutamol is quite an effective therapeutic for some forms of congenital myasthenic syndrome. ( I couldn't find mention of this in the review: it surely has a place among the content referring to clenbuterol/salbutamol in "Phase II"?)

The upshot is: this is a very nice review by experts in their fields. However, I think the authors need to decide whether it is intended to be a general historical account or one that pushes the agendas of sympathetic innervation of muscle on the one hand and the biology of AChR on the other (which may understandly be the authors' intentions given their respective research interests). I think it would be possible to do both by minor restructuring of the review: for example, to present a balanced overall but conventional update on NMJ structure, function and molecular biology to begin with but then to be quite open about identifying and justifying, in detail, the need for further research and its potential applications in selected areas that are of particular interest to the authors.

There are a few minor idiosyncracies in the use of English throughout the review but nothing worth nitpicking over. However, I did baulk at the reference in the abstract to "brilliant scientists" which is an unnecessarily florid phrase for a scientific review, in my opinion. ("If I have seen further it is by standing on the shoulders of giants", as Newton wrote: but really there are no great scientists, only the great science that many have contributed - albeit some more than others.) Also the authors use "years back" in two consecutive sentences in the abstract, which jars somewhat. An alternative, "years ago",  would work for the second occurrence.

I hope that helps. These are really only suggestions.

Author Response

Dear referee, many thanks for these detailed and sincere comments. They have definitely helped a lot to think once more about the major goals of the article. Essentially, these should be two:

First, as being the introductory text to a special issue on (>) 150 years of motor endplate research, it was thought to retrace some of the major lines of NMJ research, with a focus on the older literature, because this is more prone of being forgotten and can often not be covered by recent reviews due to space constraints. Given that a number of other colleagues will bring in their views on specialized fields for this special issue such as Schwann cells, CMS, toxins etc., we concentrated on those aspects that will most likely not be covered by the other manuscripts. Yet, it should be still a self-standing review. By reading the comments of this referee, it appears that the first mission might have been accomplished.

Second, yes, it is certainly right to say that sympathetic innervation and molecular biology of AChR are of special interest to these authors. So, we took up your suggestion and made a decision. On lines 50 ff. we state that there is a focus on sympathetic innervation and also give a rationale for this decision. Also, the title and abstract a modified to reflect these changes. Arguably, this does not make the text less unbalanced, but at least a reason for this has been mentioned now.

Concerning the suggestion to include the work Ca2+ imaging at Drosophila, we would prefer to leave it out. Of course, not because of the quality and relevance of these studies, which are excellent. However, we opted to restrict the review to vertebrate NMJs as a cholinergic synapse and putting in “a bit” of Drosophila work would likely request to add much more to keep the balance.

The work on salbutamol is actually included at lines 451 ff.

Finally, minor changes were made to the text. Thank you very much for pointing this out.